# Patterns of Traditional and Modern Uses of Wild Edible Native Plants of Chile: Challenges and Future Perspectives

**DOI:** 10.3390/plants11060744

**Published:** 2022-03-11

**Authors:** Pedro León-Lobos, Javiera Díaz-Forestier, Rodrigo Díaz, Juan L. Celis-Diez, Mauricio Diazgranados, Tiziana Ulian

**Affiliations:** 1Grupo de Especialidad en Recursos Genéticos, Centro Regional de Investigación La Platina, Instituto de Investigaciones Agropecuarias, La Pintana, Santiago CP 8831314, Chile; 2Centro Regional de Investigación e Innovación para la Sostenibilidad de la Agricultura y los Territorios Rurales, CERES, Quillota CP 2260000, Chile; javi@litre.cl; 3Grupo de Especialidad en Recursos Genéticos, Centro Regional de Investigación Quilamapu, Instituto de Investigaciones Agropecuarias, Chillan CP 3800062, Chile; rodrigo.diaz@inia.cl; 4Escuela de Agronomía, Pontificia Universidad Católica de Valparaíso, Quillota CP 2260000, Chile; juan.celis@pucv.cl; 5Royal Botanic Gardens, Kew, Welcome Trust Millennium Building, Wakehurst, Ardingly RH17 6TN, UK; m.diazgranados@kew.org (M.D.); t.ulian@kew.org (T.U.)

**Keywords:** edible plants, plant genetic resources, traditional knowledge

## Abstract

Wild Edible Plants (WEPs) still play a vital role in the subsistence of many traditional communities, while they are receiving increasing recognition in tackling food security and nutrition at the international level. This paper reviews the use patterns of native WEPs in Chile and discusses their role as future crops and sources of food products. We conducted an extensive literature review by assessing their taxonomic diversity, life forms, consumption and preparation methods, types of use (traditional and modern), and nutritional properties. We found that 330 native species were documented as food plants, which represent 7.8% of the total flora of Chile. These species belong to 196 genera and 84 families. The most diverse families are Asteraceae (34), Cactaceae (21), Fabaceae (21), Solanaceae (20) and Apiaceae (19), and the richest genera in terms of number of species are *Solanum* (9), *Ribes* (8), *Berberis* (7), *Hypochaeris* (7) and *Oxalis* (6). Perennial herbs are the predominant life form (40%), followed by shrubs (35%), trees (14%), and annual and biannual herbs (11%). Fruits (35.8%), roots (21.5%) and leaves (20.0%) are the parts of plants consumed the most. Nine different food preparation categories were identified, with ‘raw’ forming the largest group (43%), followed by ‘beverages’ (27%), ‘savoury preparations’ (27%), and ‘sweet’ (13%). Almost all native Chilean WEPs have reported traditional food uses, while only a few of them have contemporary uses, with food products mainly sold in local and specialised markets. Species’ richness, taxonomic diversity and family representation have similar patterns to those observed for the world flora and other countries where surveys have been carried out. Some Chilean native WEPs have the potential to become new crops and important sources of nutritious and healthy products in the food industry. However, there are still many gaps in knowledge about their nutritional, anti-nutritional and biochemical characteristics; future research is recommended to unveil their properties and potential uses in agriculture and the food industry.

## 1. Introduction

One of the most fundamental values of plant biodiversity for human beings is supplying the world’s food and nutrition security [1,2]. The importance of biodiversity has gradually been acquiring greater recognition in the work of international agencies such as the Food and Agriculture Organisation of the United Nations (hereafter FAO) Commission on Genetic Resources for Food and Agriculture (http://www.fao.org/cgrfa (accessed on 12 February 2022)). The benefits derived from biodiversity are also at the heart of the Convention on Biological Diversity [3] and the Global Strategy for Plant Conservation [4]. More specifically, the role it plays in food security and nutrition is proclaimed in the International Treaty on Plant Genetic Resources for Food and Agriculture [5] and the U.N. Sustainable Development Goal 2: Zero Hunger [6]. Biodiversity is fundamental to addressing the double challenge of shortage (hunger) and excess (obesity) of calories and nutrient intake that humanity is increasingly facing today, as the world’s population is expected to reach 10 billion by 2050 [7].

The practice of consuming wild edible plants (WEPs) is as old as human prehistory, and it still holds for some traditional communities today [8]. However, the relative importance and degree of dependency that humans have on WEPs varies significantly from one culture to another [9,10]. In many traditional cultures, WEPs complement staple foods to provide a balanced diet as part of the daily nutritional intake [11,12,13,14,15]. In more modern cultures such as Europe, their use is influenced by the contemporary awareness of wellbeing, health and fitness or is driven by cultural and ethical concerns [16,17,18]. In both cases, the practice of consumption of WEPs is linked to the cultural identity and deep connections between people, their land and associated traditional ecological knowledge about their surrounding natural environment and lifestyles [9,10].

Locally available WEPs can provide consumers with a more diverse range of nutritionally high-quality compounds [19,20]. They can be a source of energy, fibre and micronutrients and offer a large spectrum of phytochemicals such as phenols, tannins, flavones, terpenoids, polysaccharides, steroids, saponins and alkaloids [19,21,22,23,24]. Thus, WEPs can increase the nutrient content of poor diets [16] and produce health benefits [25,26,27]. This awareness has generated increasing interest in, and innovative uses of, WEPs as potentially exploitable sources of foodstuffs [14], functional foods, products engineered to fulfil special dietary needs, ethnic food and products with Protected Designation of Origin (PDO) that coexist with traditional uses [28].

Indigenous people in Chile, along with their occupation of the territory and interaction with their surrounding natural resources, have long been collecting and consuming a significant number of wild plants as a source of food. This traditional ecological knowledge has been documented since the Spanish conquest in the 16th Century [29,30,31,32]. However, ethnobotanical research has intensified only in the last century [33,34,35,36]. Villagran and Castro (2004) [36] gathered the most comprehensive information on the traditional uses of plants for indigenous communities from the Altiplano of northern Chile, while De Mösbach (1992) [35] completed something similar for the plant traditional knowledge of Mapuche communities of southern Chile. Díaz-Forestier et al. (2019) [37] recently compiled an inventory of the uses of the native flora of Chile by extracting uses cited in the literature until 2015. They reported that there are at least 228 native edible plants among the useful plants, representing 5% of the total flora of Chile.

However, the latter publications focused on the taxonomic distribution pattern of flora use in Chile, while previous publications have mainly produced information on medicinal plants [38,39,40,41,42]. Information on native WEPs of Chile is still fragmented [41], and methods of consumption and preparation and potential uses in the food industry have received little attention so far [43,44].

This study reviews the traditional and current knowledge of the use of native Chilean WEPs with the aim of detecting patterns of consumption and preparation according to their taxonomy diversity, life forms and types of uses. Finally, it discusses their potential contribution to nutrition and uses in the food industry.

## 2. Results

### 2.1. Taxonomic Diversity and Life Form

Our dataset contains 330 native WEPs with reported use as food. The recorded species belong to 196 genera and 84 families (see Appendix A for species descriptions). Botanical families with the highest numbers of edible species are presented in Figure 1. Asteraceae have the highest frequency with 34 WEP species, followed by Cactaceae (21), Fabaceae (21), Solanaceae (20), Apiaceae (19), Poaceae (13), Grossulariaceae (13) and Myrtaceae (13). Genera with more WEP species include *Solanum* with nine edible Crop Wild Relatives (CWRs) of tomato and potato, eight *Ribes* (currants), seven *Berberis* (barberry) and seven *Hypochaeris* species.

Perennial herbs (*n* = 130) are the predominant life form with c. 40% of the total Chilean native WEPs, including vines (*Dioscorea* spp.), epiphytes, aquatic, parasites and succulent perennial herbs (Table 1). These are followed by shrubs that are 35% of the Chilean native WEPs, including 18 succulents (Cactaceae), trees (14%) and finally annual and biannual herbs (11%). Of the total 330 species with known edible uses, 23% are endemic to Chile (*n* = 75), and 14.6% (*n* = 33) are threatened [45]. For details see Appendix A.

WEPs are sparsely and unevenly distributed across plant phylogeny, with noteworthy concentrations of species in families in the clade Asterids [Cornales (0 spp.) + Ericales (9) + Eurasterids (Lamiids (64) + Campanulids 36))], with 109 WEPs (33.5%), followed by Rosids (Fabids (49 spp.) + Malvids (37) + Vitales (0)), with 86 WEPs (26.5%). Monocots contain 48 WEPs (14.8%), while gymnosperms and Monilophytes (i.e., ferns) have only 3 WEP species each (Figure 2). The percentages are somewhat correlated to the total diversity of those clades in the Chilean flora: Asterids (40.8% of total species vs. 33.5% of WEPs; Rosids (22.1% vs. 26.5%) and monocots (21.5% vs. 14.8%). Only three of the nine species of gymnosperms are WEPs. Likewise, out of the 147 species of monilophytes, only three are WEPs. Between 40 and 70% of the WEPs in all major clades (>10 WEPs) have medicinal uses reported as well.

### 2.2. Consumption and Preparation Methods

According to our revision, the main consumed parts of Chilean native WEPs are fruits and infructescences (35.8%), roots (21.5%), leaves (20.0%), seeds (9.1%), and stems (7.0%) (Table 2). Inflorescences, exudates, bark, seedlings, and the entire plant had lower representation, less than 4% of the Chilean native WEPs. It was not possible to find information on the part of the plant used for about 10% of native WEPs. The botanical families most represented in each plant part category are similar to those reported for the total WEPs (Figure 1). Families with higher numbers of edible plant species also have higher numbers of species in each category of the part used (Table 2). For example, Asteraceae is one of the most representative families with edible leaves, stems, inflorescences, entire-plants and unspecified plant parts, as for the total Chilean native WEPs.

The most frequent form of consumption was raw (43% of species) among the nine different categories (Table 3), followed by beverages (27%), savoury preparations (27.6%) and sweet dishes (13.3%). Oils and “Other preparations” categories were below 10%. Edible native WEPs with no information about their consumption mode were around 19%. Families that are more representative in the number of species in each consumption category are generally the same ones observed for the part plant categories.

The relation between the organs consumed and the preparation forms (Table 4) showed that around 80% of the edible fruits are eaten raw, followed by sweet dishes (33%) and beverages (30%). Interestingly, there was no information about the form of consumption only for 5% of the wild edible fruits. Savoury preparation (54.9%) and raw (38%) were the primary forms of consumption of edible roots, but, compared to fruit, there was a higher percentage (38%) for which there was no information. Edible leaves and seeds showed a similar consumption pattern to roots; savoury preparation, beverages and raw were the primary forms of consumption. Leaves were preferably consumed for seasoning, and seeds as a source of oil. Most stems of WEPs had a savoury preparation (65%) and were less used. Edible flowers were consumed in savoury (30%), raw (30%) and sweet dishes (20%). However, there were no data for more than one-third of the edible flowers. Finally, over half of the species with unspecified information on the edible part were consumed as beverages, and nearly a quarter (23.5%) had no information on the manner of consumption.

Interestingly, just one plant part is consumed for over 75% of the native WEPs (Figure 3), followed by 11.7 and 2.7% of WEPs with two and three consumed plant parts. Four plant parts were consumed for just one species (0.6%), *Jubaea chilensis* (Molina) Baill., an endemic Chilean palm tree; fruits and seeds are eaten fresh and used in sweet dishes; the sap is boiled to produce syrup [47,48]; and young stems in the past were consumed as hearts of palm or “palmitos” [49].

Forty-eight percent of total native Chilean WEPs have only one reported preparation method (Figure 3). Interestingly, 21.7%, 8.4% and 3.0% native WEP are consumed in two, three and four different food preparations, respectively. Species with the highest number of preparation forms were the Chilean hazelnut (*Gevuina avellana* Molina) (*n* = 6), followed by quinoa (*Chenopodium quinoa* Willd.), chañar (*Geoffroea decorticans* (Gillies ex Hook. & Arn.) Burkart) and maqui (*Aristotelia chilensis* (Molina) Stuntz), with five forms of preparation (Appendix A).

There was an overlapping of species in nearly all categories of consumption, but mostly between raw and savoury preparations (12.3%), between raw and beverages (13%), and raw and sweet dishes (10.8%).

According to our search for modern or industrial food products, such as flour, oil, juice or pulp (available in specialised local markets, on e-commerce or in supermarkets), we found products developed only from 24 species, corresponding to 7.3% of total native Chilean WEPs (Table 5). We did not include food products for those native species widely used in the food and agriculture industry, such as *Solanum tuberosum* L., *Chenopodium quinoa*, *Pouteria lucuma* (Ruiz and Pav.) Kuntze (Sapotaceae; Lúcuma), *Physalis peruviana* L. (Solanaceae; gol-denberry), *Oxalis tuberosa* Molina (Oxalidaceae; Oca) and *Centella asiatica* (L.) Urb. (Apiaceae; Asiatic Pennywort) which are frequently found in the food markets.

## 3. Discussion

We found 330 native WEPs in Chile, 102 more species than those compiled in the previous review [37]. These species represent 7.8% of the native flora, which is similar to the percentages calculated in other countries, such as Spain (5.9%) [12], Ethiopia (6.8%) [15], Ecuador (9.2%) [50] and Mexico (9.3%) [51].

Asteraceae, Cactaceae, Fabaceae, Solanaceae and Poaceae are among the prominent families used for human food in Chile. This finding is generally consistent with a recent global study [7] and with other countries such as India [52] and Ecuador [50]. In India, Fabacaeae, Asteraceae and Poaceae are the families with most WEPs, followed by Malvaceae and Rosaceae [52]. Fabaceae and Solanaceae are also among the families with more representative species within the Ecuadorian flora [50]. In the Mediterranean area, Asteraceae, Lamiaceae, and Apiaceae are among the six most representative families [53].

It is important to highlight that Chenopodiaceae and Solanaceae are among the families containing the most main crops, such as *Chenopodium quinoa* (quinoa) and *Solanum tuberosum*. (potato), respectively; while Rosaceae and Grossulariaceae contain most CWRs; e.g., *Fragaria* (strawberry), *Rubus* (raspberry) and *Ribes* (currant). Some WEPs can also be highlighted because of their taxonomic uniqueness. For example, *Lardizabala biternata* Ruiz and Pav. (Lardizabalaceae) and *Gomortega keule* (Molina) Baill. (Gomortegaceae) belong to monotypic and endemic families of Chile [54]. Some species of Araucariaceae, Proteaceae and Elaeocarpaceae are WEPs that are culturally important for the local Mapuche communities, such as *Araucaria araucana* (Molina) K. Koch, *Gevuina avellana* and *Aristotelia chilensis* [35], respectively.

Asterids, which include Ericales, Lamiids and Campanulids, was the richest clade, containing some of the most important families (i.e., Apiaceae (e.g., wild parsley, *Osmorhiza berteroi* DC.), Asteraceae (Tupinambo, *Helianthus tuberosus* L.), Ericaceae (Chaura, *Gaultheria mucronata* (L.f.) Hook. and Arn.), Lamiaceae (tree mint, *Clinopodium chilense* (Benth.) Govaerts) and Solanaceae (Pichi Romero, *Fabiana imbricata* Ruiz and Pav.)). The second largest clade was Rosids, with families such as Fabaceae (Chilean mesquite, *Prosopis chilensis* (Molina) Stuntz) and Rosaceae (wild strawberry, *Fragaria chiloensis* (L.) Mill.). Smaller clades, but still remarkable, were the monocots, including Poaceae (pasto del perro, *Bromus catharticus* Vahl)) and Bromeliaceae (chupón, *Greigia sphacelata* (Ruiz and Pav.) Regel); and Saxifragales with Cactaceae (quisco, *Leucostele chiloensis* (Colla) Schlumpb.)). Ulian et al. (2020) [7] highlighted the relevance of some of these clades in terms of species richness (e.g., orders Asterales, Fabales, Rosales, Poales). However, some families such as Cactaceae (with 21 WEPs out of 104 total species) and Grossulariaceae (with 8 WEPs out of 9 total species) have increased significance in the Chilean flora.

Fruits, roots and vegetables (including leaves and stems) are the plant parts most consumed, which is similar to the Ecuadorian flora [50], the South American region of Gran Chaco [55], Ethiopia [15], Nepal [56] and Canada [57]. However, this is in contrast with results from ethnobotanical studies carried out in several European countries, such as the Czech Republic [58], Bulgaria [59], the Mediterranean region [12,53,60], and further afield in China [61] and India [52], where leaves and whole plants (green vegetables) are the parts that are the most frequently consumed in term of number of species in their respective floras.

Most of the Chilean native WEPs have only one edible part recorded. This finding is consistent with previous reports for the floras of Great Britain, New Guinea, Panama and other countries worldwide [62]. Patterns and level of overlap between the main modes of consumption were consistent with those reported for the wild edible vegetables and fruits of Spain [63].

Most of the Chilean WEPs were reported in early ethnobotanical studies and chronicles from naturalists. We only found two species with modern industrial uses: *Quillaja saponaria* Molina (Quillajaceae; quillay), whose saponins are used as natural emulsifiers, foaming agents in beverages and production of low-cholesterol foods [64] and *Tara spinosa* (Molina) Britton and Rose (Fabaceae; Tara) whose gum is extracted from seeds and used in commercial galactomannans in the development of edible film, or as a stabiliser, thickener, coating, emulsifier, adsorbent or gelling agent [65]. There are also new forms of consumption reported for some traditional edible species, such as for the Chilean bellflower, copihüe (*Lapageria rosea* Ruiz and Pav.), whose fruits had been traditionally consumed by the Mapuche people [35], while today its flowers are consumed in salads, chutneys and dressings.

The traditional use of WEPs has vastly decreased due to the erosion of traditional knowledge [9,17,18,66] driven by globalisation, modernisation (i.e., changes in food systems) and market integration [67]. Thus, the large proportion of native Chilean WEPs reported in ethnobotanical studies does not necessarily mean that these species are currently consumed [68]. Unfortunately, our analysis did not allow us to explore which species are still traditionally used today. However, we believe that the use of several species has been either forgotten or is very rare, such as *Madia sativa* Molina and *M. chilensis* (Nutt.) Reiche, whose seeds were used as a source of oil by the Mapuche communities [35]. Similarly, the seeds of *Bromus berteroanus* Colla, *B. catharticus* Vahl and *B. mango* E. Desv were used to make flour and bread before the arrival of the Spanish colonisers [35]. Notably, the old cereal *B. mango* E. Desv. is already extinct in the wild [45], according to IUCN criteria and categories.

Many edible plants are also used for medicinal purposes according to several studies [25,69,70,71]. In our search, 56% of the total native WEPs are also reported as medicinal, however this is comparatively lower than that reported in Ulian et al., 2020 [7] at the global scale (70%).

This survey found a relatively high percentage (Table 4) of Chilean native WEPs traditionally consumed as vegetables and roots, seeds and even flowers, however these species are scarcely valued and studied. Native edible vegetables and flowers can be a potential area for development and a source of economic income for local communities, by promoting local, ethnic and boutique cuisine [17] and as part of gastronomic tourism, similar to some European countries [12,60]. The domestication and cultivation of wild edible vegetables and flowers could support the sustainable use and conservation of native Chilean WEPs [72].

There is also high potential to produce healthy food products based on native WEPs of Chile. The real importance of these foods can be assessed in terms of their specific compounds, their quality, their quantity, and their nutritional, health and culinary properties [23]. Likewise, the toxicity, allergenic and anti-nutritional properties for humans needs to be assessed [73]. A full evaluation of these species would allow diversification of the offer of healthy and/or functional products and gourmet flavours and provide new business opportunities for small farmers.

In recent decades Chile has made an effort to become a global agro-economic leader. The launch of ‘The Food Transformation Programme’ is a strategic initiative to reinforce the sustainable use of the country’s biodiversity for economic development [74]. The key actions of this programme include the provision of (a) high-quality raw material, (b) developing healthy food products and (c) producing natural ingredients and additives for the food industry [75]. Despite this effort, there are no specific mentions of native WEPs as sources of natural ingredients for new food products. The updated Chilean National Biodiversity Strategy and Action Plans [76] does not include explicit measures for ensuring the sustainable use and conservation of WEPs. Addressing native Chilean WEPs is critically important to compete in the marketplaces dominated by a few commodity crops [7]. Consequently, an Integrated Approach for Conserving and Sustainably Using native Chilean WEPs is urgently required [8]. Critical information on their chemical composition, sociocultural aspects, biology and ecology are needed [7,8] to develop and strengthen policies supporting their conservation and sustainable use [8]. It is also relevant in the planning and implementation process to consider the role of local communities that traditionally use the WEPs [18] and assess the impact on people’s diet and livelihoods [7].

Finally, most of the analysed information on native Chilean WEPs came from the traditional knowledge gathered from ancestral communities, which in turn has been compiled by the first naturalist [29,30,31,32] and in various ethnobotanical studies (e.g., [34,35,36,77]). Consequently, ethnobotany plays a significant role in systematising and preserving the traditional knowledge associated with the WEPs [78]. It also contributes to sustainable development and preserving the biocultural diversity [79,80]. However, an appropriate and effective governance mechanism would need to be put in place to safeguard the rights of indigenous people and local communities, to manage sustainably and to benefit from the use of WEPs [81].

## 4. Material and Methods

### 4.1. Study Area

Chile is a long, narrow country situated along the western coast of South America (Figure 4). It extends approximately 2700 miles (4300 km) from its boundary with Peru, at latitude 17°30′ S, to the tip of South America at Cape Horn, latitude 56° S. Almost the entire eastern border is the continental divide of the Andes Range. 

Chile has a highly diverse climate, from the hyper-arid desert in the tropical north to the cold subantarctic southern tip. Central Chile has a Mediterranean-type climate; further south the climate is humid and cold temperate. Before the Spanish arrived the Chilean territory was inhabited by at least 13 indigenous groups [82]. At present, nine indigenous groups are recognised by law; the Mapuche in the south are the largest group, followed by the Aymara in the extreme north and Rapanui in Easter Island [83].

### 4.2. Literature Search

Initial data were obtained from the database on the uses of Chilean native flora (https://usosplantasnativas.cl/chile/ (accessed on 12 February 2022)) based on 718 documents published until July 2015. Sources of information included chronicles, natural history reports, botanical and anthropological textbooks, theses and scientific papers searched through the Web of Science database (WOS) and Google Scholar. We expanded the literature review, (WOS and Google Scholar), until July 2020 and included information on the other uses of WEP published in Chile and neighbouring countries. We used a list of relevant keywords, including: Chile, Argentina, Perú, Bolivia, ethnobotany, edible plant, fruits, leaves, beverages, seasoning, in combination with the binomial scientific name. Specific strings were used for the queries, for example: [Argentina + ethnobotany], [binomial name + edible], [binomial name + beverages] and so on. All queries were made both in Spanish and English.

The taxonomy of the species was updated based on the catalogue of the flora of Chile (http://catalogoplantas.udec.cl/ (accessed on 12 February 2022)) by Rodríguez et al., (2018) [54] and by reconciling their names against the Plants of the World Online taxonomic backbone (POWO; http://www.plantsoftheworldonline.org/ (accessed on 12 February 2022)). The nomenclature follows the International Plant Name Index [84] and the World Checklist of Vascular Plants v.5.0 [85], with which POWO is directly linked. The reconciliation of scientific names was carried out in R version 4.1.0, using the packages plyr version 1.8.5 [86], rgdal version 1.4-8 [87], dplyr version 1.0.6 [88] and doBy version 4.6-3 [89].

### 4.3. Taxonomic Diversity, Origin and Life Form

Life forms and origins of Chilean native plants were obtained from the catalogue of the flora of Chile [54]. To infer the distribution of WEPs across plant phylogeny, we mapped their presence/absence at the family-level onto a modified phylogenetic tree of vascular plants [90], indicating which families had at least one species in the Chilean flora. For this analysis we used the packages ggtree v.3.0.2 [91], ggplot2 v.3.3.5 [92], and treeio 3.13 [93] in R v.4.1.0.

### 4.4. Consumption and Preparation Methods of WEP

We considered all species naturally occurring in Chilean territory and used as food and beverages as native wild edible plants (WEPs). According to the Economic Botany Data Collection standard (EBDCS) [46], plant uses were classified as traditional, modern industrial or just anecdotical or possible uses. Following EBDCS [46], Chilean WEPs were also classified according to (i) the part of the plant used: unspecified parts, entire plants, unspecified aerial parts, seedlings/germinated seeds, galls, stems, bark, leaves, inflorescences, infructescence, seeds, roots, exudates; (ii) the form of preparation: raw, beverages, savoury preparations, sweet dishes, seasoning, cereal/starch base preparations, oils, other preparations and unspecified (see the description of categories in Table 2 and Table 3).

## 5. Concluding Remarks

The flora of Chile has a high number of native WEPs, some of which were domesticated by the original peoples of south-western South America before the Spanish conquest (Peru, Bolivia, and Chile). In contrast, others were transformed or are the progenitors of important main crops that are part of the world food base, such as potatoes, quinoa, and Chilean strawberries. However, many more WEPs have the potential to become new crops and sources of new industrial food products in the future.

Innovation based on Chilean WEPs requires strengthening both the biochemical studies to characterise the nutritional content of species in detail and agriculture research for the domestication of species and the development of breeding programmes enabling the creation of new crops.

However, in the face of future social, cultural, economic, environmental and climatic change scenarios, the food development of native Chilean WEPs must be responsibly supported by sustainable consumption and food innovation, which require access to plant genetic material and the cultivation of these plants in line with international access and benefit-sharing agreements and the creation of sustainable value chains.

## Figures and Tables

**Figure 1 plants-11-00744-f001:**
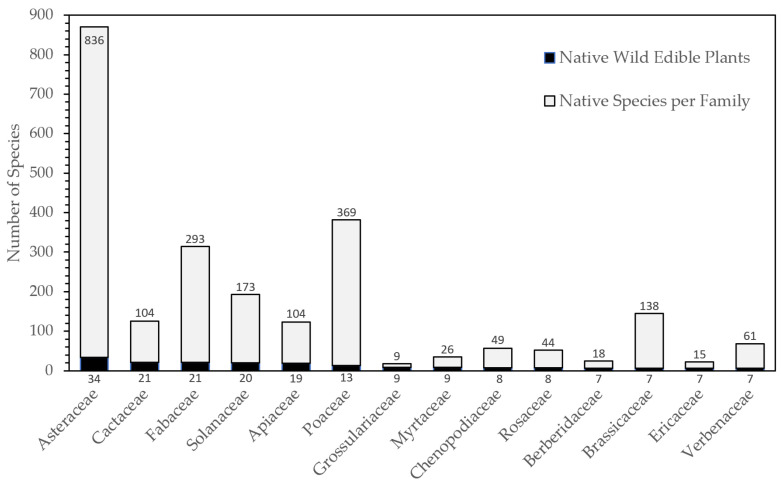
Taxonomic families with the largest numbers of native WEP species.

**Figure 2 plants-11-00744-f002:**
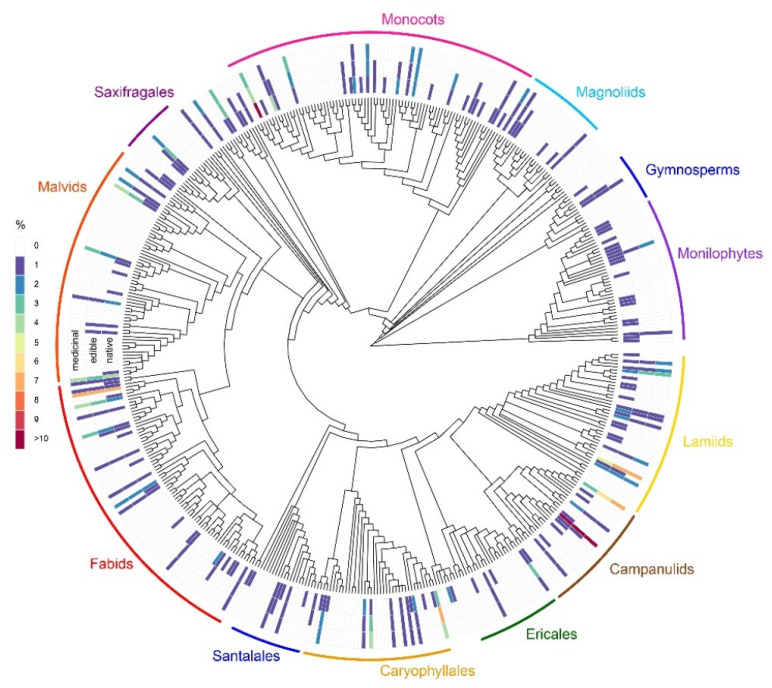
Distribution of native, edible and medicinal plants across the phylogeny of plant families. Native: inner ring of blocks; edible: middle ring; medicinal: outer ring. Colours of blocks: percentage of native, edible and medicinal species of each family, compared to the total families. *N* = 330 species. A high-resolution diagram with plant family names included is shown in Appendix A.

**Figure 3 plants-11-00744-f003:**
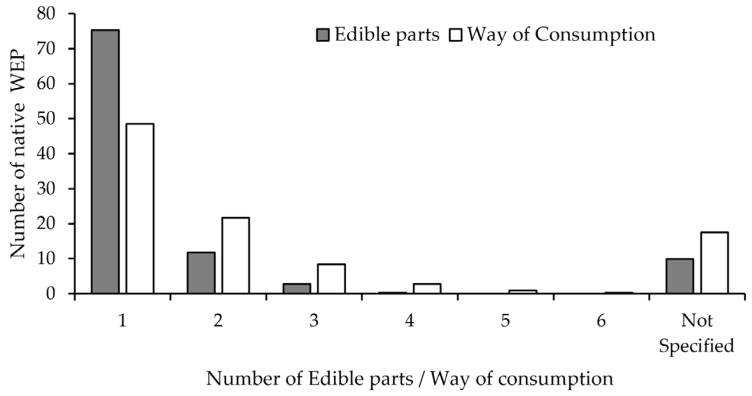
Frequency distribution of the number of edible parts and the way of consumption for the native WEP of Chile. *N* = 330 species.

**Figure 4 plants-11-00744-f004:**
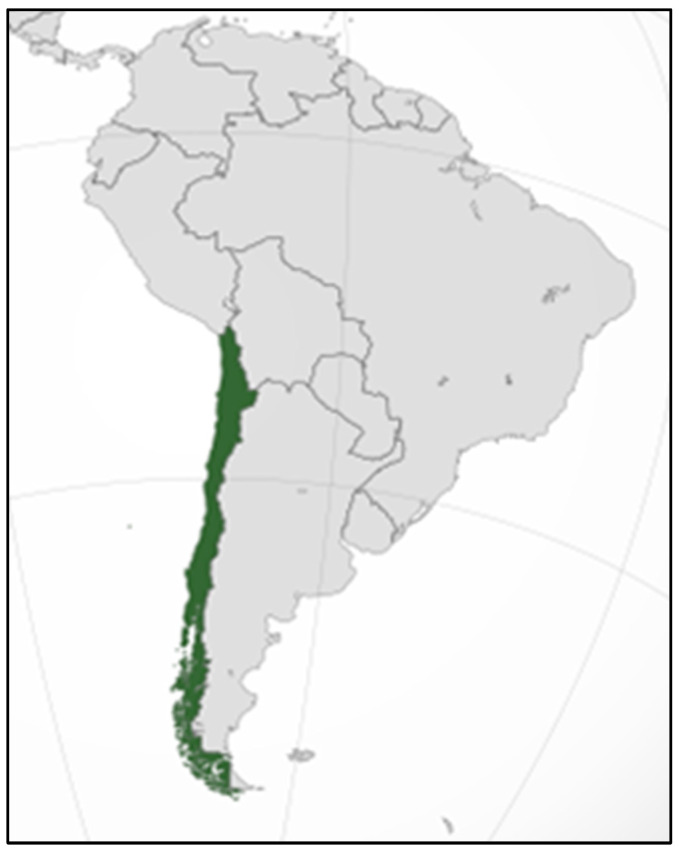
Geographic localization of Chilean territory.

**Table 1 plants-11-00744-t001:** Life forms of native WEPs of Chile.

Life Form	Species Number	(%)
Tree	36	
Succulent tree	3	
Shrub/small tree	8	
**Total trees**	**47**	**14.2**
Shrub	90	
Parasitic shrub	2	
Succulent shrub	18	
Climbing shrub	6	
**Total shrubs**	**116**	**35.2**
Perennial herb	120	
Aquatic perennial herb	2	
Epiphytic/terrestrial perennial herb	1	
Parasitic perennial herb	1	
Climbing perennial herb	5	
Succulent perennial herb	1	
**Total perennial herbs**	**130**	**39.4**
Annual herb	30	
Annual/bi-annual herb	7	
**Total herbs**	**37**	**11.2**
**Total**	**330**	

**Table 2 plants-11-00744-t002:** Number of species, percentage, number of families and families with more species (main families) according to plant parts used. Specific parts used and descriptions according to the Economic Botany Data Collection Standard (EBDCS [46]). *N* = 330.

Plant Parts Used	Description	Number of Species	%	Number of Families	Main Families
Infrutescences	Fruits, entire immature fruits, entire mature fruits, deseeded fruits, fruit pulp, fruit juice, epicarp.	118	35.8	33	Cactaceae, Grossulariaceae, Myrtaceae, Berberidaceae, Ericaceae
Roots	Debarked ‘roots’, bulbs, corms, tubers, tubercles, nodules, aerial roots, pneumatophores, rhizomes	71	21.5	25	Apiaceae, Fabaceae, Dioscoreaceae, Alstromeriaceae, Solanaceae
Leaves	Cotyledons, young leaves, old leaves, fallen leaves, leaflets, stipules, leaf blades, leaf buds, petioles	66	20.0	14	Asteraceaae, Apiaceae, Chenopodiaceae, Lamiaceae, Oxalidaceae
Seeds	Arils, entire seeds, seed hairs, seeds without testa, kernels, seed oil, seed cake, solid albumen, liquid albumen	30	9.1	11	Chenopodiaceae, Fabaceae, Poaceae, Proteaceae, Celastraceae
Stems	Plumules, leafy stems/branches, defoliated stems/branches, stolons, tendrils	23	7.0	18	Asteraceae, Rubiaceae, Oxalidaceae, Montiaceae
Inflorescences	Bracts, spathes, spadices, flowers, flower buds, peduncles, receptacles, calyces, corollas, stamens, pollen, pistils.	10	3.0	8	Asteraceae, Berberidaceae, Philesiaceae
Exudates	Sap, latex, leaf juice, gum, resin, nectar	9	2.7	4	Fabaceae, Arecaceae, Asteraceae, Nothofagaceae
Bark	Stem bark, inner bark, root bark	3	0.9	2	Rosaceae, Quillajaceae
Seedlings	Seedlings, germinated seeds	1	0.3	1	Chenopodiaceae
Entire plant	Leaves, stems, flowers and roots	4	1.2	3	Asteraceae, Brassicaceae
Unspecified aerial parts		18	5.5	4	Asteraceae, Apiaceae, Aizoaceae
Unspecified parts		32	9.7	20	Brassicaceae, Polygonaceae, Verbenaceae

**Table 3 plants-11-00744-t003:** Number of species, percentage, number of families and families with more species according to the preparations in which the plants are used. Food preparation categories and their descriptions according to the (EBDCS) [46]. *N* = 330.

Preparations	Description	Number of Species	%	Number of Families	Main Families
Raw	Unprocessed	142	43.0	44	Cactaceae, Apiaceae, Grossulariaceae, Myrtaceae, Solanaceae
Beverages	Alcoholic beverages, non-alcoholic beverages, juices, infusions/tisanes, coffee substitutes, tea substitutes	89	27.0	40	Grossulariaceae, Asteraceae, Fabaceae Anacardiaceae, Rosaceae
Savoury preparations	Soups and diverse cooked dishes (boiled, toasted, fried)	91	27.6	39	Apiaceae, Asteraceae, Fabaceae, Solanaceae, Chenopodiaceae
Sweet dishes	Confectionery, jams, jellies, syrups, ice creams	44	13.3	20	Grossulariaceae, Anacardiaceae, Berberidaceae, Ericaceae, Fabaceae
Seasoning	Condiments, relishes, chutneys, dressings	30	9.1	15	Asteraceae, Apiaceae, Verbenaceae
Cereal/starch-based preparations	Porridges, cakes pastry/shortening	27	8.2	11	Poaceae, Chenopodiaceae, Fabaceae
Oils		7	2.1	4	Asteraceae, Fabaceae, Proteaceae
Other preparations	Dehydrated, lyophilised, colorants	13	3.9	6	Cactaceae, Elaeocarpaceae, Quillajaceae
Unspecified		62	18.8	25	Asteraceae, Solanaceae, Poaceae, Dioscoreaceae

**Table 4 plants-11-00744-t004:** Number and percentages of Chilean native WEPs according to the part used and form of consumption. Species can have more than one form of consumption and part consumed.

Part Uses:	Infructescences (118)		Roots (71)		Leaves (66)		Seeds (30)		Stems (23)		Inflorescences (10)		Unspecified Parts (50)	
Consumption:	*n*	%	*n*	%	*n*	%	*n*	%	*n*	%	*n*	%	*n*	%
Raw	95	80.5	27	38.0	17	25.8	6	20.0	4	17.4	3	30.0	1	2.0
Beverages	36	30.5	3	4.2	26	39.4	8	26.7	1	4.3	1	10.0	26	52.0
Savoury preparations	6	5.1	39	54.9	33	50.0	11	36.7	15	65.2	3	30.0	4	8.0
Sweet dishes	40	33.9			1	1.5	4	13.3			2	20.0		
Seasoning	2	1.7	2	2.8	13	19.7	1	3.3					12	24.0
Cereal/starch-based preparations	8	6.8	7	9.9			2	6.7					3	6.0
Oil							6	20.0						
Other preparations	5	4.2	1	1.4	1	1.5	3	10.0	1	4.3				
Unspecified	6	5.1	27	38.0	9	13.6	3	10.0	1	4.3	4	40.0	12	24.0

**Table 5 plants-11-00744-t005:** Species with industrial edible products available in specialised markets, e-commerce or supermarkets.

Scientific Name	Species Common Name	Found in Products as
*Aristotelia chilensis* (Molina) Stuntz	maqui	Dehydrated, lyophilised, colorants, beverages, sweet dishes
*Ugni molinae* Turcz.	murta, Chilean guava	Sweet dishes, beverages
*Gevuina avellana* Molina	Chilean hazelnut	Savoury preparations, sweet dishes, cereal/starch-based preparations, oils, beverages
*Berberis microphylla* G. Forst.	calafate	Dehydrated, lyophilised, beverages, sweet dishes
*Araucaria araucana* (Molina) K. Koch	araucaria (piñones)	Cereal/starch-based preparations, savoury preparations, beverages
*Eulychnia acida* Phil.	copao (rumpa)	Beverages, sweet dishes
*Lapageria rosea* Ruiz & Pav.	copihue	Sweet dishes, seasoning (dressing)
*Geoffroea decorticans* (Gillies ex Hook. & Arn.) Burkart	chañar	Sweet dishes, beverages
*Aloysia deserticola* (Phil.) Lu-Irving & O’Leary	rika rika	Seasoning, beverages
*Drimys winteri* J.R. Forst. & G. Forst.	canelo	Seasoning
*Blechnum chilense* (Kaulf.) Mett.	helecho costilla de vaca	Pickles
*Amomyrtus luma* (Molina) D. Legrand & Kausel	luma	Beverages
*Ribes magellanicum* Poir.	zarzaparrilla	Beverages, savoury preparations
*Solanum tuberosum* L.	papa	Savoury preparations
*Jubaea chilensis (Molina)* Baill.	Chilean palm (coquitos)	Sweet dishes
*Peumus boldus* Molina	boldo	Sweet dishes, beverages, oil
*Haplopappus baylahuen* J. Remy	baylahuen	Beverages
*Buddleja globosa* Hope	matico	Beverages
*Gunnera tinctoria* (Molina) Mirb.	nalca	Sweet dishes, seasoning (chutneys)
*Prosopis alba* Griseb.	algarrobo	Sweet dishes
*Acacia caven* (Molina) Molina	espino	Sweet dishes, beverages
*Cryptocarya alba* (Molina) Looser	peumo	Sweet dishes, beverages

## Data Availability

Information and traditional knowledge on edible uses of Chilean native WEPs and related references can be found at https://usosplantasnativas.cl/chile/ (accessed on 13 of February 2022).

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
