# Peer review of "Patterns of Traditional and Modern Uses of Wild Edible Native Plants of Chile: Challenges and Future Perspectives"

_plants, 2022, doi:10.3390/plants11060744_

Round 1

Reviewer 1 Report

The manuscript submitted to me for review is a summary study of the wild edible native Chilean plants, based on the traditional and current knowledge.

This is a very well-argued and well-presented study. Undoubtedly, the manuscript fell into the area of aim and scope of the Plants journal. The study is of interest to a wide range of scientists and experts.

In this manuscript the authors here the authors collect, systematize and analyze extensive information from databases on plants of the Chilean flora, much of which is available only in Spanish.

The manuscript presents for the first time in such a way collected and structured information available to a wide range of scientists and specialists. The manuscript presents data from very precisely conducted studies on information sources: chronicles, natural history reports, botanical and anthropological textbooks, thesis, and scientific papers searched through the Web of Science database (WOS) and Google Scholar. Appropriate methods were used and representative data were obtained about taxonomic diversity, origin and life form, consumption and preparation methods of wild edible plants (WEP) and nutritional properties. On the base of obtained results authors discuss the relevance and potential contribution of native Chilean WEPs to improve food security and nutrition.

The manuscript presented to me for review, through the electronic platform, consists of one file, which includes: main text, two figures and six tables. An additional Appendix 1 is noted in the manuscript (row 135 and 221. Unfortunately, this Appendix 1 is missing in the materials submitted for review, it is probably a technical omission. It should contain very extensive and important, basic information about the article. Without it, the manuscript cannot be reviewed and evaluated.

Nevertheless, I would like to point out a few things about the available version of manuscript:

Introduction: I would suggest to the authors to expand this part a bit, so that it becomes clear how they will build on what has been known so far, what this research will contribute to. For example, a few sentences more about the subjects and results of studies from references 32-36, to highlight expected novelties.

Material and Methods: Traditionally, such articles contain little information about the study area, including a map. Given the potential readers of the article, I think that adding such information will only contribute to the informativeness and completeness of the article.

The information in the text is very well and appropriately structured and presented. Tables and figures are necessary and well selected.

The information in Figure 3. is very interesting, but the form in which it is presented is inappropriate. The main part of the text is illegible. For better information, I would recommend to the authors to think of another version of the figure.

Despite my overall very good impression of the manuscript, the review can only be completed after the missing information from Appendix 1.

Author Response

The Appendix 1 is provided as Suplementary material N°1.

In the introduction, Lines 73 to 79 more detailed information in written about what has been known in Chile on the subject of manuscript submitted.

In the Methodology, we add a paragraph with general information of Chile, the study area. A map (Figure 1) is included.

About the Figure 3, a more simplified version, without including family names is proposed. Also, high resolution version is provided as Supplementary material N°2, in which include family names.

Reviewer 2 Report

A brief summary This paper reviews the use patterns of native WEPs in Chile and discusses their role as future crops and sources of food products. The research conducted an extensive literature review and valuable analyses.

General concept comments

I find the completeness of the review topic covered, the relevance of the review topic, and the appropriateness of references, sufficient enough.

Specific comments

Since antioxidant activity is analyzed some toxicity analysis would be useful too, as there are plants like Solanum and Berberis, that contain alkaloids. Also, many Asteraceans contain sesquiterpene lactones which are also toxic (for instance Schmidt, T. J. (1999). Toxic activities of sesquiterpene lactones: structural and biochemical aspects. Curr. Org. Chem3(577-608), 4.). If the toxicity is an implication for further research then nutritional properties and antioxidant activity should be left for such future research too and removed from this paper as these evaluations are related. 

Other specific comments are made in the text body

Author Response

At the moment is not possible to conduct and exhaustive revision of references for search toxicity compounds on native wild edible plants of Chile. So, we decide to remove the parts of manuscripts related to nutritional and phenolic composition.

Reviewer 3 Report

The research article entitled “Patterns of Traditional and Modern Uses of Wild Edible Native Plants of Chile: Challenges and Future Perspectives” provided a comprehensive detailing on wild edible plants of Chile. The study is potentially interesting and it is suitable for publication upon incorporating the below comments.

Comments

  • Do authors need a dot at the end of the title of this article? I suggest authors can remove the dot.
  • In abstract (line 8): authors state “The most diverse families are in terms of WEPs are Asteraceae”, I think it should be “The most diverse families in terms of WEPs are….”. Similarly, the genus ‘Oxalis’ is not in italics, whereas the other genus in the abstract are in italics. Also, only for Oxalis the numbers in the parenthesis are indicated as species whereas in others it was not indicated.
  • Methodology (Section 2.3): It is bit confusing that authors established by saying “EBDCS” were utilized to classify plant uses. Then, all of sudden they introduce “Following Cook (1995) [55]”, it should be modified. Also, at the start of this section, “We considered as…..” I think it has some issue in its sentence structure.
  • Line-121: Galls stems?
  • Methodology (Section 2.4): I find the whole paragraphs is too complicated to understand, therefore I suggest the sentences can be simplified for better reading and understanding.
  • Figure 1: The caption in Y axis is not written in English, authors have to change the text to English.
  • Table 2: The column ‘Main families’, the families can be separated by coma. Also, in the last row 9,7, I think it should be 9.7; a similar inconsistency can be found in several places in the article, authors need to carefully rectify it.
  • Line 229: on line sale? It should be online. I suggest authors can choose words like e-commerce rather than online sale.
  • Though the article is well written, still the proficiency of English is lacking in the article and it needs to improved.

Author Response

All the minor corrections on orthography and writing were accepted.

The methodology section was corrected, and the confusion was revised and clarified. EBDCS means “Economic Botany Data Collection standard” the author of this standard is Cook, 1995. Economic Botany Data Collection Standard. Prepared for the International Working Group on Taxonomic Databases for Plant Sciences (TDWG). Royal Botanic Gardens, Kew. United Kingdom; 1995, ISBN 0947643710. ( [57]

In the Figure 1 (Actual Figure 2), the caption in Y axis is written in English.

In the Table 2 and Table 3 too, the family names were separated by coma.

The manuscript was revised by a native English speaker.

Reviewer 4 Report

The work is of interest and within the scope of the journal.
English writing seems to be generally OK, although it could be improved in some cases.

Some notes:
Figures 2 and 3 are swapped
Quite often the text in the results repeats what is in the figures or tables, which should be avoided. This happens for example with figure 2, which could be removed.
The order of tables 5 and 6 should be changed.

See also the notes placed in the pdf doc of the review.

Author Response

The order and number of the Tables Figures was revised and changed. Also, repetitions and redundance in the text mentioning figures and tables were removed.

All corrections and notes made in the original document were accepted.